# Chemotherapy-Induced Upregulation of Somatostatin Receptor-2 Increases the Uptake and Efficacy of ^177^Lu-DOTA-Octreotate in Neuroendocrine Tumor Cells

**DOI:** 10.3390/cancers13020232

**Published:** 2021-01-10

**Authors:** Rashmi G. Shah, Marine A. Merlin, Samuel Adant, Fayçal Zine-Eddine, Jean-Mathieu Beauregard, Girish M. Shah

**Affiliations:** 1Neuroscience Division, CHU de Québec Université Laval Research Center, Québec City, QC G1V 4G2, Canada; rashmi.shah@crchudequebec.ulaval.ca (R.G.S.); marine.merlin.1@ulaval.ca (M.A.M.); adant.sam@gmail.com (S.A.); faycal.zine-eddine.1@ulaval.ca (F.Z.-E.); 2Oncology Division, CHU de Québec Université Laval Research Center, Québec City, QC G1R 2J6, Canada; jean-mathieu.beauregard@chudequebec.qc.ca; 3Cancer Research Center, Université Laval, Québec City, QC G1V 0A6, Canada; 4Department of Radiology and Nuclear Medicine, Université Laval, Québec City, QC G1V 0A6, Canada; 5Department of Molecular Biology, Medical Biochemistry and Pathology, Université Laval, Québec City, QC G1V 0A6, Canada

**Keywords:** NETs: neuroendocrine tumors, SSTR: somatostatin receptors, PRRT: peptide receptor radionuclide therapy, LuTate: ^177^Lu-DOTA-octreotate, chemotherapy

## Abstract

**Simple Summary:**

The peptide receptor radionuclide therapy (PRRT) with ^177^Lu-DOTA-octreotate (LuTate) is recommended for neuroendocrine tumors (NETs) which overexpress somatostatin receptors (SSTR). A combination of LuTate with chemotherapy improves its objective response in NET patients, and here we characterized chemotherapy-induced upregulation of SSTR2 receptors as a cause for this improved response to LuTate. Using multiple NET and non-NET cell lines, we examined the SSTR2 expression for up to 7 days after exposure to drugs and its effect on LuTate uptake and cell proliferation. We report that the exposure to varying doses of chemotherapeutic drugs such as temozolomide for 24 h or 5 days results in upregulation of SSTR2 receptors between 3–7 days. This effect is more pronounced in low SSTR2 expressing BON-1 cells than in high SSTR2 expressing NCI-H727 or non-NET cancer or non-cancer cells. Thus, a properly-timed pre-treatment with low doses of chemotherapy could improve therapeutic efficacy of LuTate in NET patients.

**Abstract:**

The peptide receptor radionuclide therapy (PRRT) with ^177^Lu-DOTA-octreotate (LuTate) is recommended for different types of neuroendocrine tumors (NETs) which overexpress somatostatin receptors (SSTR). A combination with chemotherapy improves objective response to LuTate in NET patients and here we characterized chemotherapy-induced upregulation of SSTR2 receptors as a cause for this improved response to LuTate. The NET cell lines with low (BON-1) or relatively high (NCI-H727) SSTR2-expression levels, and non-NET cancer and normal cells were treated with chemotherapeutic drugs and assessed for upregulation of SSTR2. We report that an exposure to low or high doses of drugs, such as temozolomide for 24 h or 5 day results in upregulation of SSTR2 between 3–7 days, increased LuTate uptake and decreased rate of cell proliferation. This effect is at the level of SSTR2-mRNA and is more pronounced in low SSTR2 expressing BON-1 than in high SSTR2 expressing NCI-H727 or non-NET cancer or normal cells. Thus, a properly timed pre-treatment with low-dose chemotherapy could not only improve therapeutic efficacy of LuTate in NET patients who are presently eligible for PRRT, but also allow PRRT to be administered to patients with low SSTR-expressing NETs, who would otherwise not respond to this modality because of insufficient radiation delivery.

## 1. Introduction

Neuroendocrine tumors (NETs) are rare cancers derived from enterochromaffin cells of the diffuse neuroendocrine system. They are most frequently observed in the gastrointestinal tract and bronchopulmonary system [1,2,3]. Surgery is a curative option for localized NETs, but for the majority of the patients, a delayed diagnosis associated with distant metastases necessitates systemic treatment. Since NETs often overexpress somatostatin receptors (mostly SSTR2), most patients receive somatostatin analogs (SSA), such as octreotide or lanreotide, which were initially developed for palliation from the debilitating symptoms caused by hormones and bioactive substances released by functioning NETs. In the randomized phase 3 clinical trials, SSAs have also been shown to prolong progression-free survival through growth inhibition of low-grade NETs [1,4,5]. However, SSA rarely cause an objective response, i.e., tumor shrinkage [6]. After progression on SSAs, or for more aggressive NETs with high proliferative index, other systemic options include targeted biotherapies such as everolimus or sunitinib, as well as chemotherapeutic drugs, such as temozolomide (TEM), streptozotocin (STZ) and 5-fluorouracil (5-FU) or its pro-drug form, capecitabine (CAP). More recently, peptide receptor radionuclide therapy (PRRT) has emerged as another option for systemic therapy of NETs. PRRT involves use of SSA tagged with radionuclides, such as ^111^In, ^90^Y or ^177^Lu to deliver targeted therapeutic internal radiation to NET cells [7]. ^177^Lu-DOTA-octreotate (LuTate or ^177^Lu-DOTATATE; Lutathera^®^) is the preferred PRRT radiopharmaceutical to date due to its high affinity to SSTR2 and its optimal β-particle energy offering improved response rates over ^111^In-based PRRT and lesser potential for toxicity than ^90^Y-based PRRT [8]. In a randomized trial, LuTate provided longer progression-free survival and improved quality of life as compared to long-acting octreotide [9,10]. PRRT efficacy figures are also more favourable than chemo- or biotherapies [8,11,12]. The recent guidelines recommend LuTate PRRT as appropriate therapy for patients with NETs of midgut, pancreas and of unknown origin, as well as for lung-NET, paraganglioma and pheochromocytoma under specific conditions [13].

Despite being one of the most successful targeted therapies for NETs so far, PRRT needs to be improved, because complete remissions are anecdotal, and a risk of significant toxicity remains an obstacle to administer higher radioactivity to all patients [9,14]. Moreover, the current clinical decision algorithms exclude PRRT for a substantial number of NET patients whose tumors are of high grade (hence poor prognosis) or classified as having low SSTR expression, i.e., tumor uptake of ^68^Ga-DOTATATE on PET scan that is less than that by liver; and thus benefit to risk ratio is deemed to be poor for such patients [13]. Therefore, various approaches are being assessed for improving PRRT including modification in the radiopharmaceutical molecule by use of α-emitting isotopes or attaching the radiolabel to SSTR antagonists instead of agonist like octreotide/octreotate [15,16]. Another promising approach is to combine LuTate PRRT with other agents to potentiate its effect on NET via different routes, such as improved tumor perfusion of LuTate, upregulation of SSTR in tumor to increase the uptake of LuTate, inducing additional lethal or sublethal damage with other agents including chemotherapeutic drugs or inhibition of allied pathways associated with tumor’s response to LuTate therapy, as reviewed recently by us [12].

Some clinical studies have shown that therapeutic combination of LuTate with TEM and/or CAP is well tolerated with substantial tumor control in different types of NETs [17,18] and a durable objective response in pancreatic NETs [19]. However, these studies did not directly compare the advantage of combining LuTate with chemotherapy over LuTate alone, which is being addressed in an ongoing phase II clinical trial involving pancreatic and midgut NETs [20]. From clinical reports, the effect of chemotherapy on enhancing uptake of PRRT is not clear, because in one series of three patients, a significant increase in the uptake of ^68^Ga-DOTATATE was seen several months after treatment with everolimus or CAPTEM [21], whereas in another series of 10 patients treated with LuTate with or without CAPTEM, no increase in tumor uptake of LuTate occurred over a 15-day period [22].

Studies with NET and non-NET cells in vitro or in animal models have examined the influence of chemotherapy on SSTR2 expression or LuTate uptake with variable results. In four different non-NET pancreatic cancer cell lines which largely express SSTR3 and no significant levels of SSTR1 or SSTR2, treatment with various chemotherapeutic agents decreased the high-affinity binding sites for ^111^In-labeled lanreotide derivative DOTA-LAN; and increased or decreased the low-affinity binding sites after a few days [23]. In three different NET cell lines, treatment with 5-FU for 72 h resulted in none, modest or significant increase in expression of SSTR2 levels in GOT1, NCI-H727 and BON-1 cells, respectively, which could be consistently increased in all cell lines by additional treatment with epigenetic modifiers [24]. In a mouse model of NET tumor with H69 small cell lung cancer cells, a 14-day treatment with TEM did not increase SSTR2 density on cells, but increased the tumor uptake of ^111^In-octreotide or LuTate due to increased tumor perfusion of the radiopharmaceutical [25]. Together these studies reveal the potential of chemotherapy to improve therapeutic efficacy of LuTate PRRT, provided there is a better understanding of the mechanism of this action. Here, using NET cells in vitro, we show that a single dose of chemotherapeutic agents, such as TEM results in a delayed but specific upregulation of only SSTR2 among all five SSTR subtypes, which results in a significantly higher uptake of LuTate. Moreover, this benefit is more pronounced in the low SSTR2-expressing BON-1 cells than in high SSTR2 expressing NCI-H727 cells, indicating another potential clinical advantage of making patients with low SSTR2-expressing NETs eligible for PRRT.

## 2. Results

### 2.1. Co-Administration of Chemotherapeutic Drugs Increase Uptake of LuTate in Low SSTR2-Expressing BON-1 Cell

To determine the effect of chemotherapy on SSTR-mediated uptake of LuTate by NET cells, we used a pair of low and high SSTR2-expressing cells, BON-1 and NCI-H727, respectively. In these cells, we have previously shown that the extent of intracellular uptake of LuTate after a 5-day exposure was commensurate with the expression levels of SSTR2 and SSTR5 [26]. Here, we first examined whether the co-administration of different chemotherapeutic drugs could influence the uptake of LuTate by these cells. For BON-1 cells, the extent of LuTate uptake was significantly increased from 5- to 16-fold by co-administration of 200 µM TEM, 0.5 mg/mL STZ, 15 µM 5-FU or a combination of STZ+5-FU (Figure 1A, left panel). As expected, the basal level uptake of LuTate in the high SSTR-expressing NCI-H727 cells was ~5-fold greater than in BON-1 cells (Figure 1A, both panels). However, the drug-induced increase in the uptake of LuTate in these cells was a more modest but significant 1.9- to 3.4-fold (Figure 1A, right panel). Interestingly, the levels of LuTate uptake in BON-1 cells treated with TEM or STZ was 4 times more than the basal uptake of LuTate in NCI-H727 cells and comparable to that in some of the drug-treated NCI-H727 cells.

To measure the consequence of increased uptake of ^177^LuTate, we determined the proliferation rate of BON-1 and NCI-H727 cells by counting viable cells at 96 h after treatment of an identical set of cells with drugs and LuTate, exactly as described above (Figure 1B). The comparison of viable cell-counts at the start and end of the protocol revealed that the untreated BON-1 cells multiplied ~12-fold during the experimental period, and LuTate treatment slightly, but significantly suppressed it to 9-fold growth (Figure 1B, left panel). The treatment with only TEM or STZ caused a significant suppression with only 3- to 4-fold increase in the viable cell count. However, a combination treatment of cells with LuTate and either of these two drugs completely suppressed proliferation of BON-1 cells and even decreased the viable cell count indicating cytostatic and cytotoxic effect of the combination therapy. Unlike TEM or STZ, the treatment with 5-FU and 5-FU+STZ that caused a 5- to 8-fold increase in uptake of LuTate (Figure 1A, left panel) did not translate into additional suppression of growth of cells (Figure 1B, left panel). A near total suppression of growth of BON-1 caused by given doses of these two drugs may be the cause for lack of any additional cytostatic or cytotoxic effect of LuTate treatment. Unlike BON-1 cells, the NCI-H727 cells were very sensitive to growth inhibitory effects of each of these drugs per se at their given doses, and there was no significant additional cytostatic effect of LuTate on these drug-treated cells (Figure 1B, right panel). Our results indicate that a moderate growth suppressive effect of LuTate in the low SSTR-expressing BON-1 cells can be potentiated by sub-toxic doses of drugs that increase the uptake of LuTate. Therefore, we focused on BON-1 cells for further characterization of upregulation of LuTate-uptake in response to lower sub-toxic doses of drugs, with a view that this could lead to a clinically relevant benefit for patients with low SSTR-expressing tumors [12,13].

### 2.2. A Delayed Upregulation of SSTR2 in TEM-Treated BON-1 Cells

We first examined whether an increased expression of the principle somatostatin receptor SSTR2 could be one of the reasons for an increased uptake of LuTate in the TEM-treated BON-1 cells, as described above in Figure 1A. Indeed, the immunoblotting of these cells harvested at 5 days after treatment with the drugs and LuTate exhibited a robust drug-induced increase in expression of SSTR2 (Figure 2A). Next, we characterized the drug-induced upregulation of SSTR2 in BON-1 cells at multiple levels, such as the time-course of upregulation, minimum treatment time and dose of drug required for upregulation, and whether the drug and LuTate need to be administered together or sequentially. To address these questions, we treated the cells with TEM for either one or five days at 0 (mock control), 30, 100 or 300 µM doses and compared the SSTR2 expression daily from 2–7 days (Figure 2B). At all doses of TEM, the SSTR2 upregulation began from 3–4 days and peaked between 5–7 days. The upregulation was stronger at higher (100 and 300 µM) doses than at lower (30 µM) dose. Interestingly, the shorter 24 h treatment with the drug was as efficient as the continuous 5-day treatment in causing the same pattern of delayed upregulation of SSTR2 (Figure 2B). Lastly, the peak SSTR2 levels remained stable for two more days after the 5th day, irrespective of whether the drug was removed after 24 h or 5 days. Thus, TEM-induced upregulation of SSTR2 in BON-1 cells exhibited two key characteristics, namely a brief 24 h exposure to TEM was as efficient as long-term (5d) exposure, and the time-delayed upregulation of SSTR2 from 3–5 days persisted at high levels until 7 days.

### 2.3. Sequential Treatment of TEM and LuTate Also Increases Intracellular Uptake of LuTate in BON-1 Cells

In view of the 6.7 days half-life of ^177^Lu and the 5- to 7-day lag period for peak upregulation of SSTR2 after 24 h treatment with TEM, we reasoned that an optimum schedule of treatment would be to wait for about 5 days after 24 h treatment with TEM before administering LuTate to ensure that the radiolabel is delivered at the peak expression of the receptor. To test this concept, we exposed BON-1 cells to 30 or 100 µM TEM for 24 h, followed by recovery in drug-free medium up to 7 days. During the recovery period, their capacity for LuTate uptake was measured each day from 4–7 days after a brief 3 h exposure to LuTate (Figure 3A). The uptake of LuTate was 5- to 9-fold greater from 4 to 7 days in 30 or 100 µm TEM treated cells, as compared to mock-treated cells where LuTate uptake remained unchanged during this period. These results indicate that the sequential administration of TEM followed by delayed administration of LuTate also benefits from TEM-induced upregulation of SSTR2 in a 4–7 days window after TEM treatment. We confirmed using multiple dose and time exposure of BON1 cells to LuTate that a brief (3–6 h) exposure to LuTate is sufficient to measure LuTate uptake by these cells (Figure 3B). Moreover, we confirmed that ~85% of total LuTate taken up by the cells is internalized and resistant to acid-wash, whereas residual LuTate that could be eluted with acid wash was either on the SSTR2 receptors or non-specifically bound to the plasma membrane (Figure 3C).

### 2.4. TEM Treatment Specifically Upregulates mRNA of SSTR2 among Five SSTRs

In order to examine the cause for drug-induced increase in SSTR2 in BON-1 cells, we examined the abundance of mRNA for all five SSTR genes by qRT-PCR for 6 days after treatment with even lower dose (10 µM) of TEM (Figure 4A). When normalized for expression levels of either of the two of the control genes GAPDH or G6PD, only the expression of SSTR2 revealed an upregulation starting at third day and peaking around fifth day (Figure 4A). We confirmed that there was an absolute increase in copy number of SSTR2 mRNA during this period. The identical increase in the abundance of SSTR2 transcript relative to two different control genes G6PD and GAPDH, and the lack of upregulation of other SSTR in the same samples, together exclude unrelated variation in the mRNA of control or SSTR genes skewing the results. It is also noteworthy that BON-1 cells have undetectable levels of SSTR4 and higher expression of SSTR5 than SSTR2, but the treatment that increased SSTR2 did not alter levels of these low or high expressing SSTR genes, supporting the argument for selective upregulation of SSTR2. The increased transcription of SSTR2 at 4–5 days was confirmed by immunoblotting for SSTR2 in protein extracts of these cells harvested daily up to 6 days (Figure 4B). Thus, the increased transcription of SSTR2 gene is a key factor in the drug-induced upregulation of SSTR2 protein and consequent increase in the uptake of LuTate by these cells. We confirmed using 3D spheroids of BON-1 cells [26] that 10 µM TEM treatment for 24 h upregulates SSTR2 expression from 4–6 days as compared to mock-treated spheroids (Figure 4C).

### 2.5. Effect of Sequential Treatment with Drugs and LuTate on LuTate Uptake and Cell Proliferation

We examined the effect of 24 h treatment with 10 µM TEM, 10 µM 5-FU or 50 µg/mL STZ on LuTate uptake at the 5th day (Figure 5A) and cell proliferation at the 10th day, i.e., five days after administration of LuTate (Figure 5B). The uptake of LuTate was increased by 2-, 2.9- and 7.8-fold in response to TEM, 5-FU and STZ, respectively (Figure 5A). The uptake of LuTate after drug treatment was mediated by SSTR, because it was between 5–20 times stronger than the non-specific adsorption of radiolabel to the cells as determined by incubation with equivalent radioactivity from ^177^Lu-DTPA (Appendix A). In terms of suppression of proliferation rate of cells, while LuTate and TEM had some effect in reducing the growth rate of BON-1 cells, the sequential treatment with 10 µM TEM and LuTate caused a significantly greater growth-suppressive effect than that caused by either treatment alone (Figure 5B). In case of STZ, the drug alone caused a significant decrease in proliferation of cells, but addition of LuTate almost totally suppressed the growth of these cells. For 5-FU-treated cells, there was a moderate suppression of growth by the drug, which was not increased by treatment with LuTate (Figure 5B) despite 3-fold more uptake of LuTate, as noted above in Figure 5A. We noted that this is largely due to cells being pushed into senescence by 5-FU treatment, which could be reversed by treatment with Rapamycin, 5 h before treatment with 5-FU (Appendix A). Collectively, these results show that combined or sequential treatment with sub-toxic doses of TEM and STZ stimulates expression of SSTR2 in low SSTR2-expressing BON-1 cells, which results in increased LuTate-uptake and improved therapeutic response that is significantly higher than that observed with the drug or LuTate alone. Moreover, in case of 5-FU, the drug-induced senescence appears to be a major factor in reducing the impact of uptake of LuTate on cell growth and this could be alleviated by use of senescence inhibitor.

### 2.6. Comparison of TEM-Induced Uptake of LuTate by NET and Other Normal or Cancerous Cells

Finally, we compared the pancreatic and lung NET cells (BON-1 and NCI-H727, respectively) with other cancer cell lines, such as U2OS, MCF7, HeLa and H187, and normal cells (NL-20 and HEK) for alterations in their capacity for uptake of LuTate at the 5th day after 24 h treatment with 30 µM TEM (Figure 6). We confirmed that LuTate uptake in low SSTR2-expressing BON-1 cells increased by about 5.9-fold in response to TEM treatment, as compared to 1.6-fold in high SSTR2 expressing NCI-H727 cells. The HEK cells derived from kidney, which have basal expression of SSTR2 similar to BON-1 cells, responded to TEM treatment with only 1.6-fold increase in LuTate uptake, which is 4-fold less than the response of BON-1 cells. The basal LuTate uptake in HeLa, NL-20 and MCF-7 was very low and despite TEM-induced increase in uptake ranged from 1.1- to 17.7-fold, none of these TEM-treated cells took up more than 5% of the uptake by TEM-treated BON-1 cells. The low basal uptake of LuTate in untreated H187 small cell lung cancer cells increased by ~2-fold after TEM treatment, but it was still 10-times less than seen in TEM-treated BON-1 cells. Lastly, the untreated U2OS osteosarcoma cells, which had basal LuTate uptake that was almost 5-fold more than BON-1, exhibited a modest 1.6-fold increase in LuTate uptake after TEM treatment that was similar to that seen is high SSTR2-expressing NCI-H727 cells. These results suggest that TEM treatment induced upregulation of SSTR2 and LuTate uptake in NET cells may not be observed in other non-target organs irrespective of whether these organs had a tendency for low or high levels of uptake of LuTate before chemotherapy.

## 3. Discussion

We observed that the exposure of low SSTR2-expressing BON-1 cells to a chemotherapeutic agent TEM at varying doses (from 10 to 300 µM) for 24 h or 5-days results in a significant upregulation of SSTR2 between 3–7 days. The SSTR2 upregulation is associated with a corresponding increase in the uptake of LuTate and a consequent suppression of proliferative capacity of these cells. The TEM treatment specifically increases the abundance of mRNA for SSTR2 and not the other four SSTR subtypes. The TEM-induced upregulation of SSTR2 is also seen in the 3D-spheroids of BON-1 cells. The higher SSTR2-expressing NCI-H727 cells also respond to TEM treatment with a significant upregulation of SSTR2 and an increased uptake of LuTate, although to a more modest extent as compared to low SSTR2-expressing BON-1 cells. Both the cell lines also respond to other drugs, such as STZ and 5-FU with an increased uptake of LuTate. Finally, as compared to BON-1 cells, the TEM-induced upregulation was much less pronounced in non-target cells irrespective of their high or low basal level-expression of SSTR2, suggesting less risk for higher toxicity in non-target organs.

Together our results show that combining sub-toxic doses of chemotherapy followed by a targeted administration of LuTate at peak SSTR2 expression can result in better therapeutic efficacy of PRRT while minimizing the side-effects chemotherapy if it were to be given at its prescribed high dose-regimes. Previous clinical studies have reported that a combination of chemotherapy (CAP and/or TEM) with LuTate provides an improved objective response while being well-tolerated by the patients, albeit with higher hematological toxicity rates than reported for LuTate alone [17,18,19]. In the prospective clinical trials by Claringbold et al. [17,19], LuTate was administered on the 5th day after start of CAP treatment for 14 days during which TEM was given in the last five days in each cycle. After 4 cycles given at 8-week intervals, the follow up of patients revealed an objective and durable response. In a retrospective study, Yordanova et al. [18], reported that 15 NET patients for whom PRRT or chemotherapy alone had earlier failed to control the disease, an objective response was obtained when LuTate was administered either one or 10 days after TEM or CAPTEM treatment. Based on our results, we argue that the timing of LuTate administration during the peak SSTR2 upregulation after lower dose chemotherapeutic regimes could result in better outcome of the combination therapy with much reduced hematological toxicity as compared to previous studies.

Our observation that chemotherapy-induced upregulation of SSTR2 is more pronounced in low SSTR2-expressing BON-1 cells than in higher SSTR2 expressing NCI-H727 cells is supported by two clinical studies. It has been reported that when patients with different types of metastatic NETs with low SSTR positivity (or Krenning score) were given six cycles of CAPTEM, their tumors showed an improved uptake of ^68^Ga-DOTATATE at the end of several months of chemo treatment [27]. On the other hand, patients with metastatic NETs with high SSTR positivity did not get any additional uptake of LuTate when it was administered 9 days after start of a 14-day CAP regime in which TEM was given for the last 5 days [22]. Interestingly, in a mouse model of NET tumor with high SSTR2-expressing NCI-H69 small cell lung cancer cells, Bison et al. [25] showed that chemotherapy can augment LuTate uptake by the tumor not via increased SSTR2 but via improved tumor perfusion of LuTate. In our cellular studies in vitro with high SSTR2-expressing NCI-H727 cells, tumor perfusion is not a confounding factor, but we nonetheless observed a modest but significant increase in the receptor-mediated uptake of LuTate after chemotherapy. Together, our results along with these reports suggest that chemotherapy could enhance therapeutic effect of LuTate by improved SSTR2-mediated uptake and via better perfusion of the radiopharmaceutical in the tumors with low or high SSTR2 status, which need to be directly assessed in preclinical and randomized controlled clinical studies.

The inhibitors of epigenetic modifiers such as DNA methyl transferases and histone deacetylases also improve transcription of SSTR and somatostatin in NET and non-NET cancers [28]. Specifically in BON-1 cells, Veenstra et al. [29] showed that a 7-day treatment with DNMT inhibitor 5-aza cytidine and HDAC-inhibitor Valproic acid increased the expression of mRNA of SSTR2, but not any other SSTR subtypes, a pattern of selective upregulation of SSTR2 similar to that seen by us in these cells with chemotherapeutic drugs. Teleman et al. [30] also observed upregulation of SSTR2 at mRNA and protein levels in BON-1 cells 3 days after treatment with DNMT inhibitor decitabine or HDAC inhibitor tacedinaline or a combination of both the drugs. The alteration in expression levels of SSTR2 has also been reported in different NET cell lines in response to 5-FU combined with decitabine or tacedinaline [24]. This study used BON-1, NCI-H727 as well as GOT1 and QGP1 NET cell lines to show that 5-FU pre-treatment upregulates SSTR2 at 3 days in BON-1 cells, and it could be enhanced further by combination with epigenetic modifiers, whereas this effect was less evident in high SSTR2-expressing NCI-H727 and GOT1 cells. This study did not examine the time course of SSTR2 upregulation or its beneficial effect on LuTate therapy, but did show increased sensitivity of all the cell lines to external ionizing radiation after these treatments [24]. Thus, epigenetic drug-induced upregulation of SSTR2 has the potential for becoming additional therapeutic option for improving PRRT of NET cancers. Transcriptional upregulation of SSTR2 has also been shown with physiological factors such as insulin and growth hormones in liver cells of rainbow trout [31] and it has been correlated with octreotide-induced overexpression of miR-16-p in INS-1 rat insulinoma NET cell line [32]. Unlike these other approaches discussed above for increasing SSTR2 expression to improve PRRT, our study suggests that we could use chemotherapeutic agents, which are already approved for NET treatment, to achieve exactly the same end-result of upregulation of SSTR2. Moreover, we show that SSTR2 upregulation and higher uptake of LuTate can be achieved with a single sub-toxic dose of chemotherapy; and therefore, higher therapeutic efficacy of PRRT by LuTate can be achieved without using the previously examined full dose-regimes of chemotherapy that cause incidental toxicity.

## 4. Materials and Methods

### 4.1. Chemicals

Majority of the chemicals and fine chemicals were from MilliporeSigma (Oakville, ON, Canada), and cell culture-related products were from Life technologies-ThermoFisher (St-Laurent, QC, Canada). Nitrocellulose ECL membrane was from Amersham (Oakville, ON, Canada) and Immobilon western chemiluminescent HRP substrate (WBKLS0500), STZ and 5-FU were from MilliporeSigma (Oakville, ON, Canada). TEM and rapamycin were from Enzo Life Sciences (Ann Arbor, MI, USA).

### 4.2. Radiopharmaceuticals

^177^Lu-DOTA-octreotate radiolabeling was performed as described [33]. ^177^LuCl_3_ was obtained from IDB Holland BV, and [DOTA^0^,Tyr^3^]-octreotate was a gift from the Erasmus Medical Center (Rotterdam, The Netherlands). Radiochemical purity of ^177^Lu-octreotate was >97%, and specific activities ranged from 22,015 to 85,100 MBq/µmole in different batches.

### 4.3. Cells, Chemotherapy and LuTate Treatment, Measurement of LuTate Uptake and Viability

The BON-1, established from a human pancreatic carcinoid tumor [34], was maintained as described [35]. The bronchopulmonary NET cells NCI-H727 (CRL-5815), U2OS, MCF7, HeLa, NL20, HEK and H187 cells were from ATCC and maintained as per their specifications. In general, cells were seeded in triplicate at 10,000 cells/cm^2^ for viable cell count and LuTate uptake studies, and another triplicate for survival studies. For studies requiring additional immunoblotting of SSTR2, a separate set of plates were seeded, since this antibody probing worked best with scrapped and not trypsinized cells. Viable cell counts were carried out by Trypan blue (GIBCO) dye exclusion assay with trypsinized cells.

For co-administration of drugs and LuTate for 5 days (Figure 1), BON-1 and NCI-H727 cells were seeded, as above and treated after 2 days with 3.7 MBq (100 µCi) ^177^Lu-octreotate alone or with 200 µM TEM, 25 µM 5-FU or 0.5 mg/mL STZ or a combination of 5-FU and STZ. After 5 days, cells were washed thrice with phosphate buffered saline (PBS) and trypsinized for removal of non-specifically bound LuTate. The viable cell count was carried out as above. Cells were spun down and pellet was suspended in 200 µl PBS, mixed with 5 mL of scintillating liquid for measuring total cellular uptake of radioactivity by liquid scintillation β-counter (Coulter). A separate set of cells were processed on the 5th day for immunoblotting of SSTR2 (Figure 2A). For determining the effect of single and combination treatment on cell survival, a separate set of cells were treated with drug or LuTate alone or with the combination of drugs, as described above for 5 days, followed by removal of LuTate and drug on the 5th day and allowed to recover in fresh medium for 4 more days prior to cell count by trypan blue viability assay. The fold increase in viable cell count over the entire period from seeding to harvesting was determined for each group.

For a time-course of TEM-induced SSTR2 upregulation (Figure 2B), cells were treated with specified doses of TMZ (or mock controls) for either 24 h or 5 days. After treatment, cells were allowed to recover in fresh medium without the drugs for up to 7th day. On each specified day, mock and TMZ treated cells were harvested and processed for immunoblotting of SSTR2.

For time-course of TEM-induced LuTate uptake in sequential treatment studies (Figure 3A, Figure 5A and Figure 6), BON-1 or other specified cells seeded as above, were treated with specified doses of TMZ or mock treated for 24 h followed by recovery in fresh medium for specified number of days up to 7 days. On the specified day, 3.7 MBq LuTate was added in the medium for 3–6 h as specified and cells were harvested for cell count and total cellular uptake of LuTate as described above. To distinguish between internalized and membrane-bound LuTate (Figure 3B), cells were exposed to specified amount of LuTate (63–630 kBq) and processed for measuring total uptake of radiolabel by suspending the trypsinized cells in 0.5 mL PBS containing 20 mM sodium acetate pH 5.2, and incubated at 37 °C for 10 min. The cells were spun to collect supernatant as acid-wash (membrane-bound) count, and cell pellet was processed to measure internalized LuTate activity. The LuTate uptake was expressed as cpm/10^6^ cells or per mg protein, and is expressed as mean ± S.E. of 3–6 replicates for each sample. To measure cell viability after sequential treatment with various drugs and LuTate (Figure 5B), a separate set of plates were processed exactly as above for the drug treatment followed by LuTate after 4 days. After removal of LuTate, the cells were allowed to recover for 4 days and processed for measuring viable cell count. For TEM-induced upregulation of SSTR2 in BON-1 spheroids (Figure 4C), the 3D spheroids were prepared as described earlier [26], treated on the 6th day with TEM for 24 h followed by recovery in fresh medium for 6 days. The 6–12 spheroids were harvested at each day from 4–6 day, pooled and processed for immunoblotting of SSTR2.

### 4.4. Immunoblotting and Antibodies

The cells were harvested by scraping, washed with PBS, suspended in Laemmli buffer and sonicated with a microtip at 45% power for 20 sec (Fisher Sonic Dismbrator). Protein concentration was estimated by Bradford and 10–20 µg of protein extracts were resolved on 6–15% gradient SDS-PAGE, transferred to nitrocellulose membrane and probed with specific antibodies. Ponceau S staining was used as the loading control. All Western blotting data were repeated 2–3 times from the extracts derived from two independent experiments, and results are displayed from one of the replicate experiments with similar results. The chemiluminescence signal was captured on ChemiGenius under non-saturating conditions. The antibodies used for immunoblotting were SSTR2 (1:1000, Abcam, Cambridge, MA, USA) and secondary HRP-conjugated antibodies (1:1250, Jackson Immunoresearch Labs, Westgrove, PA, USA).

### 4.5. RT-PCR for SSTR1-5

Quantitative Real-Time PCR measurements were performed by the Gene Expression Platform of CHU de Québec Research Center by methods compliant with MIQE guidelines [36]. In brief, monolayers were rinsed with PBS before addition of Trizol (Life Technologies, ThermoFisher, St-Laurent, QC, Canada) to the dishes. Total RNA was extracted with RNeasy mini kit on-column DNase (Qiagen, Montréal, QC, Canada), and concentration was measured using a NanoDrop ND-1000 Spectrophotometer (NanoDrop Technologies, Wilmington, DE, USA). Its quality was assayed on an Agilent BioAnalyzer 2100 (Agilent Technologies, Santa Clara, CA, USA). First-strand cDNA was synthesized at 25 °C for 10 min, 50 °C for 20 min, inactivation at 80 °C for 10 min followed by purification on PCR purification kit (Qiagen, Montréal, QC, Canada). cDNA corresponding to 20 ng of total RNA was used to perform fluorescent-based Realtime PCR quantification using the LightCycler 480 (Roche Diagnostics, Laval, QC, Canada). Reagent LightCycler 480 SYBRGreen I Master (Roche Diagnostics, Laval, QC, Canada) was used with 2% DMSO. The conditions for PCR reactions were: 45 cycles, denaturation at 98 °C for 10 s, annealing at 60–63 °C for 10 s, elongation at 72 °C for 14 s and then 74 °C for 5 s (reading). Normalization was performed using the reference genes glucose-6-phosphate dehydrogenase (G6PD) and glyceraldehyde-3-phosphate dehydrogenase (GAPDH). Relative expression of the target gene (difference between control and TMZ treated samples for that gene) was divided by relative expression of reference gene (difference between control and TMZ treated samples for reference gene) to obtain relative quantification ratios for each time point [37]. The forward/reverse primers (5′ to 3′) used were:SSTR1: GGGTGCTATCGCTGCTCGTCATCC/AGCCCACCAGCCAGCGTTGAGSSTR2: ACCCCATCAAGTCGGCCAAGT/TCCGGAGCCCAGCATATATCATSSTR3: GTGGCGCTCTTCGTGCTCTGC/CCACCAGGAAGTAGAGCCCAAAGAAGSSTR4: CAGGCGCTCGGAGAAGAAAATC/TGGCATCAAGGCTGGTCACGASSTR5: GAAGGTGACGCGCATGGTGTT/AGAGGATGACCACGAAGAAGTAGAGGG6PD: GATGTCCCCTGTCCCACCAACTCTG/GCAGGGCATTGAGGTTGGGAGGAPDH: GGCTCTCCAGAACATCATCCCT/ACGCCTGCTTCACCACCTTCTT

Human 3β-hydroxysteroid dehydrogenase/δ5-δ4-isomerase (3β-HSD) gene intron was used a negative control to verify the absence of genomic DNA.

## 5. Conclusions

We have shown that a short, low-dose chemotherapy pre-treatment of NET cells prior to PRRT can significantly upregulate SSTR2 after a few days, particularly in cells having a low baseline expression thereof, resulting in enhanced uptake of LuTate and greater therapeutic efficacy. This represents an appealing approach to enable and potentiate PRRT for patients with low SSTR-expressing NETs, who would otherwise neither be eligible nor respond to this modality because of insufficient radiation delivery. Such a limited chemotherapy exposure would not be anticipated to add significant toxicity to the well-tolerated PRRT.

## Figures and Tables

**Figure 1 cancers-13-00232-f001:**
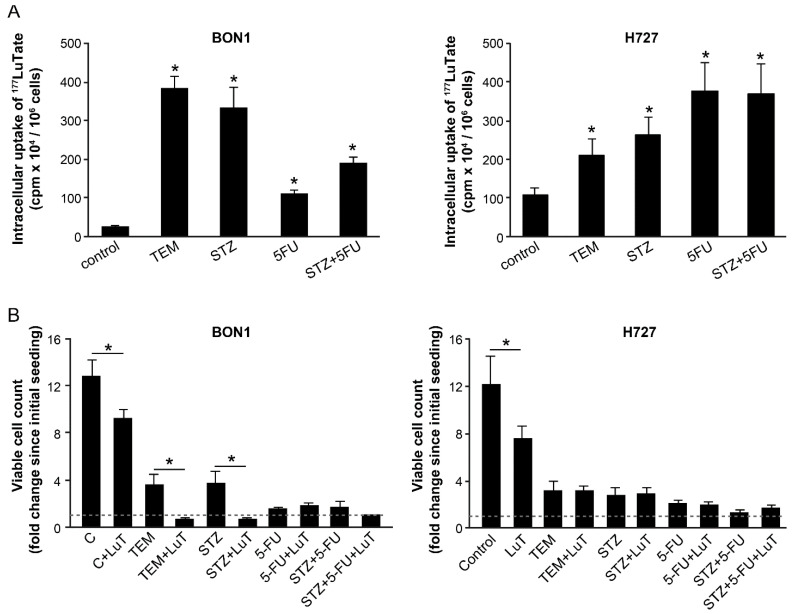
Co-administration with various chemotherapeutic drugs improves uptake of LuTate in BON-1 and NCI-H727 cells with more cytostatic effect in BON-1 cells. (**A**) Improved LuTate uptake after chemotherapy. BON-1 and NCI-H727 cells were treated for five days with LuTate alone or with 200 µM TEM, 25 µM 5-FU or 0.5 mg/mL STZ or a combination of 5-FU and STZ. The total cellular uptake of LuTate was measured on the 5th day. (**B**) A separate set of plates were treated for five days as described for panel A, washed, allowed to recover in fresh medium for 4 days prior to viable cell count, which is presented as fold increase over initial viable count, as indicated by dashed line at 1 on *Y*-axis. The data derived from two separate experiments, each with a minimum of triplicate independent dishes for each data point are represented as Mean ± SE (*n* = 6). The Student’s t-test was performed and * indicates significant difference with *p* value below 0.05.

**Figure 2 cancers-13-00232-f002:**
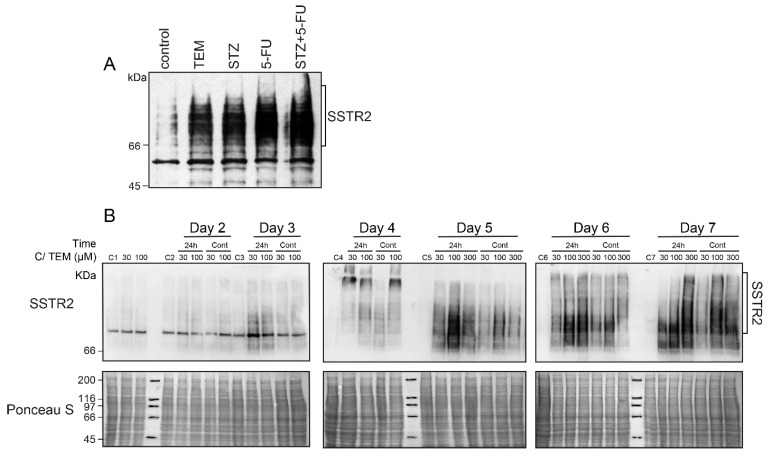
Effect of time and dose of TEM treatment on the time course of upregulation of SSTR2 in BON-1 cells. (**A**) Chemotherapy-induced upregulation of SSTR2 at five days in BON-1 cells. BON-1 cells treated with different drugs and LuTate for five days, as described in Figure 1A, were separately processed for immunoblotting for SSTR2. (**B**) TEM treatment time and dose-response on time-course of SSTR2 upregulation. BON-1 cells treated for 24 h (24 h lanes) or continuously for 5 days (Cont lanes) with specified doses of TEM or mock-treated controls (C-lanes) were harvested over 7 days for assessment of SSTR2 expression by immunoblotting. Ponceau S staining is shown as loading control. The blots shown here represent identical results obtained in at least two independent experiments for all the conditions.

**Figure 3 cancers-13-00232-f003:**
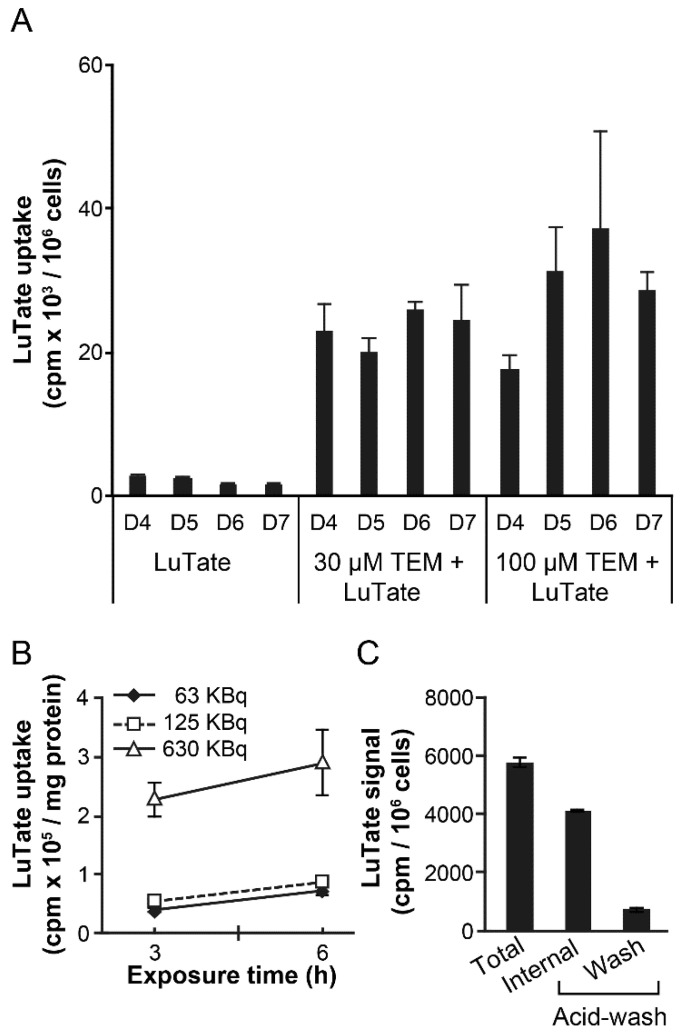
TEM-induced increase in uptake of LuTate by BON-1 cells. (**A**) LuTate uptake by TEM-treated BON-1 cells. BON-1 cells treated for 24 h with 30 or 100 µM TEM (or mock) and allowed to recover in fresh medium for 7 more days. The cells were exposed for 3h to LuTate on days 4–7 after TEM treatment, and harvested for measuring LuTate uptake and viable cell count. Data derived from triplicate samples of four independent experiments is represented as mean ± SE (**B**) Dose and time-dependent uptake of LuTate by BON-1 cells. The BON-1 cells were incubated for 3 or 6 h with specified doses of LuTate and assessed for total cellular uptake of radiolabel. (**C**) Distribution of total LuTate uptake as internalized in the cell versus plasma membrane-bound activity. BON-1 cells treated for 3 h with 125 KBq LuTate were processed for measuring total, internalized (acid wash resistant) and membrane bound (acid-wash removable) LuTate. For panels B and C, the data derived from two experiments, each with triplicate dishes are represented as Mean ± SE (*n* = 6).

**Figure 4 cancers-13-00232-f004:**
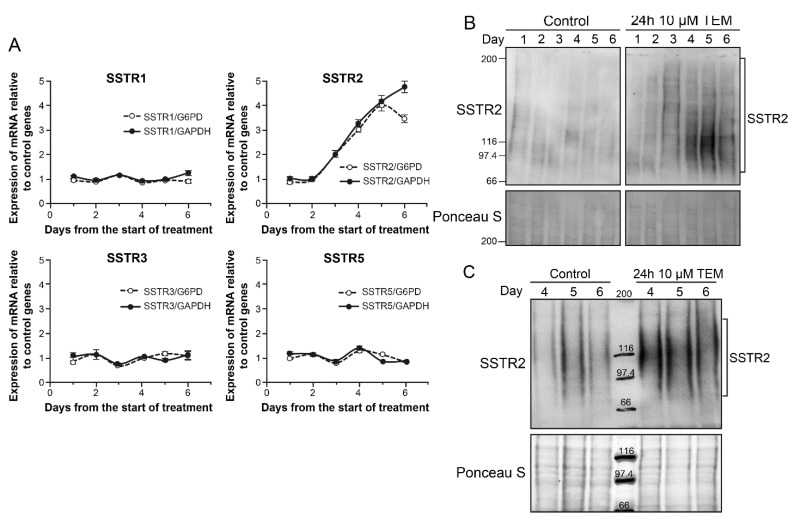
TEM-induced upregulation of SSTR2 in BON-1 cells and spheroids. (**A**) TEM-induced selective upregulation of mRNA for SSTR2 among five SSTR subtypes in BON-1 cells. BON-1 cells were treated with 10 µM TEM or mock-treated as controls, harvested at specified day for RNA extraction and subjected to qRT-PCR for measurement of abundance of mRNA of SSTR1-5. The data are expressed as relative abundance of copy number of the mRNA of given SSTR gene in TMZ treated cells as compared to mock control after normalization with the abundance of mRNA of two independent control genes G6PD and GAPDH in the respective samples. Each data point is presented as Mean ± SD from two independent experiments in which each sample was assessed in triplicate by qRT-PCR. The signal for SSTR4 was below detection limit in BON-1 cells. (**B**) TEM-induced upregulation of SSTR2 protein in BON-1 cells. BON-1 cells treated with 10 µM TEM, as described for panel A, were harvested on specified days and processed for immunoblotting of SSTR2. The blot represents identical results obtained in three independent experiments. (**C**) TEM-induced upregulation of SSTR2 protein in BON-1 spheroids. BON-1 spheroids were treated on the 6th day with TEM for 24 h followed by recovery in fresh medium for 6 days. The 6–12 spheroids were harvested at each day from 4–6 days post-TEM treatment, pooled and processed for immunoblotting of SSTR2.

**Figure 5 cancers-13-00232-f005:**
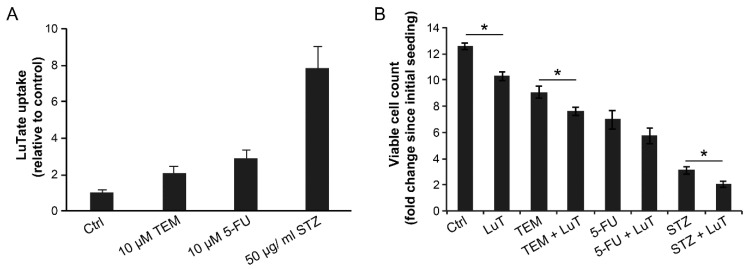
Improved LuTate uptake and growth suppression in BON-1 cells treated in sequence with chemotherapeutic drugs and LuTate. (**A**) LuTate uptake: BON-1 cells treated for 24 h with specified dose of three different drugs (or control) were allowed to recover in fresh medium for 4 days before exposure to LuTate for 4 h followed by processing for total cellular uptake of LuTate. The cells were seeded in triplicate and data derived from three independent experiments is presented as Mean ± SE. All three chemotherapy treatment groups had a statistically significant difference in LuTate uptake as compared to mock treated control. (**B**) Cell proliferation: An independent set of cells were treated with same regime but were allowed to recover for five days in fresh medium after removal of LuTate, and assessed for the fold increase in viable cell count to determine the rate of proliferation after initial seeding. The data derived from triplicates from two experiments are expressed as Mean ± SE. The * refers to a statistically significant difference (*p* < 0.05).

**Figure 6 cancers-13-00232-f006:**
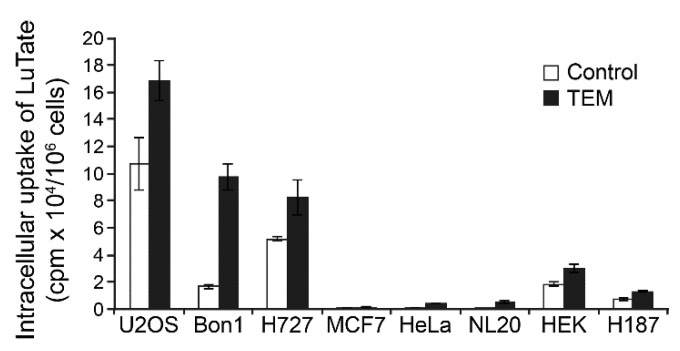
Variable response of NET, other cancers and non-cancer cell lines to TEM-induced changes in LuTate uptake. The TEM-induced changes in uptake of LuTate by NET cell lines BON-1 and NCI-H727 was compared with other cancer cell lines, such as U2OS (osteosarcoma), HeLa (cervical cancer), MCF7 (breast cancer) and H187 (small cell lung cancer), as well as with two normal cell lines NL-20 (normal bronchial epithelial), and HEK (human embryonic kidney). The cells seeded in quadruplicates were treated for 24 h with 30 µM TEM followed by recovery in fresh medium for five days. Cells were exposed for 4 h to LuTate followed by cell count and measurement of LuTate uptake per million cells, expressed as Mean ± SD.

## Data Availability

Not applicable.

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
