# Peer review of "Chemotherapy-Induced Upregulation of Somatostatin Receptor-2 Increases the Uptake and Efficacy of 177Lu-DOTA-Octreotate in Neuroendocrine Tumor Cells"

_cancers, 2021, doi:10.3390/cancers13020232_

Round 1
Reviewer 1 Report
The study is aimend to show that the use of chemotherapy increases the expression of SSTRs in NET cell lines.
Results are very interesting, but conclusions are too optimistic. It is true that some studies have suggested a good response to chemotherpy + PRRT in NET patients, but only study on patients, and a randomized controlled clinical trials (PRRT alone vs. CHT + PRRT) can draw definitive conclusions.
These considerations need to be added to discussion.
Author Response
Response to Reviewer 1:
(1) Thank you for your supportive comments.
(2) Our conclusion is optimistic but also realistic in the suggestion that our strong in vitro data supports studies to validate our approach in preclinical animal models and randomized controlled clinical trials (revised manuscript: Discussion line # 362).
(3) We agree that previous clinical studies of chemo and PRRT lacked all relevant controls, and that only a randomized clinical trial with all controls will resolve this issue: This was clearly stated in the Introduction lines 87-88 (Pavlokis et al. 2020, reference #20).
Reviewer 2 Report
The paper is interesting. In particular, it focuses on a currently unmet need: improving the PRRT outcome with Lutathera. We usually prefer human primary cell culture systems and hope you can reproduce your results in this context in the future.
I appreciated the suggestion comes from your data to activate Phase II study with Temozolomide in neoadjuvant therapy before PRRT.
Author Response
Response to reviewer 2:
1) Thank you for your positive assessment of our work that will improve PRRT outcome with Lutathera.
2) The work with primary cell lines is in our future plans for this project.
3) We are eagerly looking forward to such phase 2 clinical trials, whether conducted by us or any other groups, to rapidly translate our concept from bench to bedside for the benefit of NET patients.
Reviewer 3 Report
The work is interesting and quite well done. It addresses a topic of great and current interest such as the combined effect of chemotherapy and PRRT
Recent retrospective studies have shown a good response of combination therapy (CAP-TEM and PRRT) in patients with NET G2-G3 in terms of OS and PFS. The data is still in conflict; cases of objective response obtained after combined therapy (where PPRT or chemotherapy alone had failed) and studies that did not show significant differences between chemotherapy alone and the association with PRRT are described.
It is particularly interesting the possibility of evaluating the SSTR2 expression in the improvement of the response to LuTATE.
However, the methodology used to detect SSTR2 upregulation (qRT-PCR) is not perfectly clear, as is the mechanism by which only SSTR2 is upregulated.
Questions:
- Regarding the mRNA expression of SSTR2 in BON1-cells, was an absolute quantization performed? Or is it a relative increase in SSTR2 due to the temozolomide-induced lack of undifferentiated cells?- For a better evaluation of a real quantitative upregulation of SSTR2 did you used a Digital PCR?
- If true absolute and non-relative upregulation of SSTR2 is confirmed, do you have any explanation for the failure to upregulate the other somatostatin receptors?
- How do you explain the more pronounced effect of temozolomide (on up regulation of SST2, increased Lu Tate uptake and decrease cell proliferation) in low SSTR2 line cells despite high SSTR2 line cells ?
- What could be the clinical impact of these findings? which type of patients would benefit most from such combined treatment?
- In the study, temozolomide was given greater importance, however streptozotocin showed equally interesting results; why was it overlooked in subsequent investigations?
Author Response
Response to reviewer 3:
Response to general comments and suggestions:
-Thank you for your appreciation of our work.
- We agree that previous clinical studies of chemo and PRRT lacked all relevant controls, and that only a randomized clinical trial with all controls will resolve this issue: This was clearly stated in the Introduction lines 86-88 (Pavlokis et al. 2020, reference #20)
-Thank you for appreciating our suggestion that assessing SSTR expression (ideally by 68Ga-DOTA-Octreotate scan) in patients for several days after administration of low dose chemotherapy is the ideal way to optimize the time of administration of PRRT. The importance of timing after chemo for PRRT administration has been emphasized in the manuscript in results and discussion and in the simple summary (lines 25-26) and abstract (line 38).
Responses to Specific questions and methodology
1) Quantitative Real-time PCR methodology: The qRT-PCR was carried out by a professionally managed Genomics platform of the research centre using methodology that is compliant with MIQE guidelines, as stated in the Materials and Methods lines 461-62 (Ref 36).
2) Selective upregulation of SSTR2 and not other SSTR?
Thank you for seeking clarification of this point. A succinct version of the response given below is now added in the results section for Fig 4 (lines 222-229 and in the legend to Fig 4A-lines 238-42. Here are the responses:
(i) Yes, there was absolute increase in the abundance of SSTR2 mRNA. We specifically used mock-treated control cells for each day for comparison with TMZ treated cells on the same day.
(ii) The identical results with an increased abundance of SSTR2 transcript relative to two different control genes G6PD and GAPDH, and lack of upregulation of other SSTR, together exclude unrelated variation in the control genes or other SSTR genes skewing the final results. It is also noteworthy that BON1 cells have undetectable levels of SSTR4 and expression levels of SSTR5 that are higher than SSTR2, but the treatment that increased the abundance of SSTR2 did not alter levels of low (SSTR4) or high (SSTR5)-expressing SSTR genes, supporting the argument for selective upregulation of SSTR2.
(iii) Lastly, we have two independent proof (other than qRT-PCR) for upregulation of SSTR2, namely immunoblots for SSTR2 and increased uptake of LuTate by chemo- treated cells, as shown in numerous data panels in the results.
3) Mechanism for upregulation SSTR 2
We have discussed (last paragraph: lines 364 onwards) the SSTR2 upregulation with different types of agents and even specific upregulation of SSTR2 over other SSTR in some models. This indicates either a direct or indirect control over SSTR2 gene expression by different agents. Detailed studies to identify the mechanism of upregulation of SSTR2 is a point of our interest too, but it is beyond the scope of this manuscript that is focused on demonstrating improved efficacy of PRRT with prior chemotherapy treatment
Other SSTR are upregulated in different models, for example upregulation of SSTR3 in chemo-treated non-NET pancreatic cancer cells has been described in the Introduction-line 94 onwards (Fueger et al. ref# 23). However, since this non-NET model does not express SSTR2, which is key receptor for PRRT, we are not in a position to judge the relevance of lack of upregulation of other SSTR receptors in our NET model for PRRT.
4) Why low SSTR2 expressing cells had better therapeutic response than high SSTR2 expressing cells?
The low SSTR2 expressing cells exhibited a more robust upregulation of SSTR2 than high SSTR2 expressing cells, with a resultant higher uptake of LuTate and better therapeutic response. In addition, the higher SSTR2 expressing cells may already have saturating levels of SSTR2 and scope for improved LuTate uptake is thus limited. In addition, the high SSTR2 expressing cells also demonstrate a relatively modest increase in SSTR2 levels in response to chemotherapy. Hence therapeutic impact of chemo+PRRT is more pronounced in cells with low SSTR2 than in the cells with high SSTR2
5) Clinical impact of our study and which type of patients would benefit?
We have emphasized the advantage of using our proposed approach to provide maximum benefit to low SSTR2 patients who are currently ineligible for PRRT; and this has been stated in the summary and conclusion. We have explicitly stated this in the abstract (lines 39-41) and in conclusion (lines 492-495).
Nonetheless, we do not exclude that high SSTR2 patients, already eligible for PRRT, would not get at least some therapeutic benefit by proper timing of the PRRT treatment after low dose chemo-induced modest upregulation of SSTR2.
6) Why streptozotocin results, despite as promising as temozolomide, not followed up in the study?
We focused on temozolomide (TEM) because as TEM and CAPTEM are routinely administered chemotherapeutic regimes for NET patients. Therefore, results from previous clinical and in vitro studies with similar model would allow us to readily compare our results with the existing literature, and offer a faster route to clinical translation of our results.
We believe that STZ-PRRT approach would also be beneficial, but will have to be validated in animal models and in clinical trials.
We thank the reviewer for excellent comments and suggestions for future lines of enquiries in this project.